# IIFL: Implicit Interactive Fleet Learning from Heterogeneous Human Supervisors

**Gaurav Datta**[*1]**, Ryan Hoque**[*1]**, Anrui Gu**[1]**, Eugen Solowjow**[2]**, Ken Goldberg**[1]

[1]AUTOLab at UC Berkeley    [2]Siemens Research Lab

**Abstract:** Imitation learning has been applied to a range of robotic tasks, but can struggle when robots encounter edge cases that are not represented in the training data (i.e., distribution shift). Interactive fleet learning (IFL) mitigates distribution shift by allowing robots to access remote human supervisors during task execution and learn from them over time, but different supervisors may demonstrate the task in different ways. Recent work proposes Implicit Behavior Cloning (IBC), which is able to represent multimodal demonstrations using energy-based models (EBMs). In this work, we propose Implicit Interactive Fleet Learning (IIFL), an algorithm that builds on IBC for interactive imitation learning from multiple heterogeneous human supervisors. A key insight in IIFL is a novel approach for uncertainty quantification in EBMs using Jeffreys divergence. While IIFL is more computationally expensive than explicit methods, results suggest that IIFL achieves a $2.8\times$ higher success rate in simulation experiments and a $4.5\times$ higher return on human effort in a physical block pushing task over (Explicit) IFL, IBC, and other baselines.

## 1   Introduction

Imitation learning (IL), the paradigm of learning from human demonstrations and feedback, has been applied to diverse tasks such as autonomous driving [1, 2, 3], robot-assisted surgery [4, 5], and deformable object manipulation [6, 7, 8]. The most common IL algorithm is behavior cloning (BC) [2], where the robot policy is derived via supervised machine learning on an offline set of human task demonstrations. Since BC can suffer from distribution shift between the states visited by the human and those visited by the robot, interactive IL (IIL) algorithms including DAgger [9] and variants [10, 11, 12] iteratively improve the robot policy with corrective human interventions during robot task execution. These algorithms are typically designed for the single-robot, single-human setting; *interactive fleet learning* (IFL) [13] extends IIL to multiple robots and multiple human supervisors. However, learning from multiple humans can be unreliable as the data is often multimodal.

Training data is *multimodal* when the same state is paired with multiple (correct) action labels: $\{(s, a_i), (s, a_j), \dots\}, a_i \neq a_j$. Almost all robot tasks such as grasping, navigation, motion planning, and manipulation can be performed in multiple equally correct ways; as a result, almost all demonstration data has some degree of multimodality. Multimodality is especially severe when learning from different human supervisors with varying preferences and proficiency, as they demonstrate the same task in different ways [14]. Multimodality can also occur in the demonstrations of one individual human who may make mistakes, become more proficient at the task over time, or execute a different valid action when subsequently encountering the same state [14, 15].

Florence et al. [16] propose Implicit Behavior Cloning (IBC), an IL algorithm that trains an energy-based model (EBM) [17] to represent state-action mappings *implicitly* rather than explicitly. While this makes model training and inference more computationally expensive (Section 6), implicit mod-

---

[*]Equal contribution.

7th Conference on Robot Learning (CoRL 2023), Atlanta, USA.

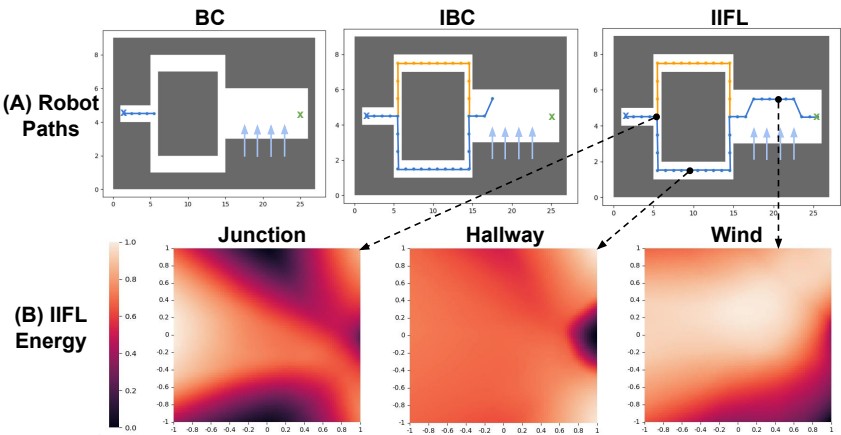

Figure 1: In the 2D navigation experiments from Section 5.1, the robot must navigate from the blue X marker on the left to the green X marker on the right, where the robot can go either above or below the rectangular grey obstacle and continue through a section subject to upward wind forces (blue arrows) that shift commanded motions upward. **(A) Robot Trajectories:** After training on 100 demonstrations of the two paths around the obstacle, pure behavior cloning cannot make progress past the fork due to multimodal demonstrations, while Implicit Behavior Cloning cannot overcome the distribution shift due to wind in the $+y$ direction at execution time (denoted in light blue). IIFL reaches the goal by handling both multimodality and distribution shift. **(B) Implicit Interactive Fleet Learning Energy:** We display normalized IIFL energy distributions from representative states in the trajectory. Lower energy (darker) indicates a more optimal action, and the $x$ and $y$ axes are the 2D action deltas $\hat{a}$ that the robot can execute (which can be mapped directly onto the corresponding $1\times1$ cell in the maze). At the junction point, both upward and downward actions attain low energy; in a straight hallway, the rightmost actions attain low energy; in the windy area, actions toward the lower right corner (making progress toward the goal while fighting the wind) attain low energy.

els can represent multiple actions for each state. This property allows them to handle both single-human multimodality and multi-human heterogeneity, as they are indistinguishable from a data-centric perspective. However, IBC suffers from the same distribution shift as (Explicit) BC.

In this paper we combine implicit models with interactive fleet learning to facilitate interactive learning from multiple humans. See Figure 1 for intuition. As existing IFL algorithms rely on estimates of epistemic uncertainty like the output variance among an ensemble of networks, which are incompatible with implicit models (Section 4.3), we propose a new technique for estimating the epistemic uncertainty in EBMs using Jeffreys divergence [18].

This paper makes the following contributions: (1) Implicit Interactive Fleet Learning (IIFL), the first IIL algorithm to use implicit policies, (2) a novel metric for estimating uncertainty in energy-based models, (3) simulation experiments with a fleet of 100 robots and 10 heterogeneous algorithmic supervisors, (4) physical experiments with a fleet of 4 robots and 2 heterogeneous human supervisors. Open-source Python code is available at https://github.com/BerkeleyAutomation/IIFL.

## 2 Preliminaries and Related Work

### 2.1 Interactive Imitation Learning

Learning from an offline set of human task demonstrations with behavior cloning (i.e., supervised learning) is an intuitive and effective way to train a robot control policy [19, 2, 6, 1]. However, behavior cloning can suffer from distribution shift [9], as compounding approximation errors and real-world data distributions (e.g., variable lighting in a warehouse) can lead the robot to visit states that were not visited by the human. To mitigate distribution shift, Ross et al. [9] propose dataset aggregation (DAgger), an IIL algorithm which collects online action labels on states visited by the robot during task execution and iteratively improves the robot policy. Since DAgger can request excessive queries to a human supervisor, several IIL algorithms seek to reduce human burden by intermittently ceding control to the human during robot execution based on some switching criteria

[11, 10, 20]. Human-gated IIL [11, 21, 22] has the human decide when to take and cede control, while robot-gated IIL [23, 10, 12, 20] has the robot autonomously decide. Hoque et al. [13] propose Interactive Fleet Learning (IFL), which generalizes robot-gated IIL to multiple robots supervised by multiple humans. In this work, we consider the IFL setting.

Sun et al. [24] propose a method for interactive imitation learning from heterogeneous experts, but their method is not based on implicit policies and is limited to autonomous driving applications. Gandhi et al. [25] also interactively learn from multiple experts and propose actively soliciting the human supervisors to provide demonstrations that are compatible with the current data. However, this prevents the robot from learning alternative modes and requires the human supervisors to comply with suggestions, which may not occur due to human suboptimality, fatigue, or obstinacy [26].

## 2.2 Robot Learning from Multimodal Data

Learning from multimodal demonstrations is an active challenge in machine learning and robotics. A mixture density network [27] is a popular approach that fits a (typically Gaussian) mixture model to the data, but it requires setting a parameter for how many modes to fit, which may not be known a priori. When actions can be represented as pixels in an image (e.g., pick points), a Fully Convolutional Network [28] can be applied to learning pixelwise multimodality [8, 29]. Shafiullah et al. [30] propose Behavior Transformers, a technique that applies the multi-token prediction of Transformer neural networks [31] to imitation learning. Other Transformer-based policies report similar benefits for multimodal data [32, 33]; however, these approaches require action discretization to cast behavior prediction as next-token prediction. In a very recent paper, Chi et al. [34] introduce diffusion policies, an application of diffusion models [35] to imitation learning from multimodal data.

Florence et al. [16] propose implicit behavior cloning, a technique that trains a conditional energy-based model [17] and is found to outperform (explicit) BC and mixture density networks in their experiments. As opposed to explicit models that take the form $\pi : \mathcal{S} \rightarrow \mathcal{A}$, implicit models take the form of a scalar-valued function $E : \mathcal{S} \times \mathcal{A} \rightarrow \mathbb{R}$; the action is an input rather than an output of the model. To sample an action from the policy, instead of evaluating the explicit model $\hat{a} = \pi(s)$, the implicit model must perform optimization over $E$ conditioned on state $s$:

$$\hat{a} = \arg\min_{a \in \mathcal{A}} E(s, a) \tag{1}$$

In this work, we combine IBC with IFL to mitigate the effects of both distribution shift and multimodality. To our knowledge, we are the first to extend implicit policies to interactive IL.

## 2.3 Jeffreys Divergence

The Jeffreys divergence [18] is a statistical measure of the distance between two probability distributions and is a symmetric version of the Kullback-Leibler (KL) divergence:

$$D_J(P\|Q) = D_{KL}(P\|Q) + D_{KL}(Q\|P).$$

The KL divergence is widely used in machine learning algorithms, most commonly in variational autoencoders [36] and generative adversarial networks [37]. It has also been used for dimensionality reduction [38], information bottlenecks [39], and policy gradient methods for reinforcement learning [40, 41]. The Jensen-Shannon divergence [42] is another symmetric KL divergence that sums the KL divergences of both distributions against the mixture of the two, but neither the Jensen-Shannon nor the asymmetric KL divergences have the structural properties that make Jeffreys divergence amenable to our setting (Section 4.3). Nielsen [43] derives a proposition similar to Identity 1 (Section 4.3) with Jeffreys divergence for exponential families but does not apply it to energy-based models. To our knowledge, IIFL is the first algorithm to use Jeffreys divergence for uncertainty estimation in energy-based models, exploiting its structural properties for fast computation.

# 3 Problem Statement

We consider the interactive fleet learning (IFL) setting proposed by Hoque et al. [13]. A fleet of $N$ robots operate in parallel independent Markov Decision Processes (MDPs) that are identical apart from their initial state distributions. The robots can query a set of $M < N$ human supervisors with action space $\mathcal{A}_H = \mathcal{A} \cup \{R\}$, where $a \in \mathcal{A}$ is teleoperation in the action space of the robots and $R$ is a "hard reset" that physically resets a robot in a failure state (e.g., a delivery robot tipped over on its side). As in [13], we assume that (1) the robots share policy $\pi_{\theta_t} : \mathcal{S} \to \mathcal{A}$, (2) the MDP timesteps are synchronous across robots, and (3) each human can only help one robot at a time. However, unlike the original IFL formulation [13], we do *not* assume that the human supervisors are homogeneous; instead, each human $i$ may have a unique policy $\pi_H^i : \mathcal{S} \to \mathcal{A}_H$. Furthermore, each $\pi_H^i$ may itself be nondeterministic and multimodal, but is assumed to be optimal or nearly optimal.

An IFL supervisor allocation algorithm is a policy $\omega$ that determines the assignment $\boldsymbol{\alpha}^t$ of humans to robots at time $t$, with no more than one human per robot and one robot per human at a time:

$$\omega : (\mathbf{s}^t, \pi_{\theta_t}, \cdot) \mapsto \boldsymbol{\alpha}^t \in \{0,1\}^{N \times M} \quad \text{s.t.} \quad \sum_{j=1}^M \boldsymbol{\alpha}_{ij}^t \leq 1 \text{ and } \sum_{i=1}^N \boldsymbol{\alpha}_{ij}^t \leq 1 \quad \forall i, j. \tag{2}$$

The allocation policy $\omega$ in IFL must be autonomously determined with robot-gated criteria [10, 12] rather than human-gated criteria [11, 21, 22] in order to scale to large ratios of $N$ to $M$. The IFL objective is to find an $\omega$ that maximizes return on human effort (ROHE), defined as the average performance of the robot fleet normalized by the amount of human effort required [13]:

$$\max_{\omega \in \Omega} \mathbb{E}_{\tau \sim p_{\omega, \theta_0}(\tau)} \left[ \frac{M}{N} \cdot \frac{\sum_{t=0}^T \bar{r}(\mathbf{s}^t, \mathbf{a}^t)}{1 + \sum_{t=0}^T \|\omega(\mathbf{s}^t, \pi_{\theta_t}, \boldsymbol{\alpha}^{t-1}, \mathbf{x}^t)\|_F^2} \right] \tag{3}$$

where $\|\cdot\|_F$ is the Frobenius norm, $T$ is the amount of time the fleet operates (rather than an individual episode horizon), and $\theta_0$ are the initial parameters of $\pi_{\theta_t}$.

# 4 Approach

## 4.1 Preliminaries: Implicit Models

We build on Implicit Behavior Cloning [16]. IBC seeks to learn a conditional energy-based model $E : \mathcal{S} \times \mathcal{A} \to \mathbb{R}$, where $E(s, a)$ is the scalar "energy" for action $a$ conditioned on state $s$. Lower energy indicates a higher correspondence between $s$ and $a$. The energy function defines a multimodal probability distribution $\pi$ of action $a$ conditioned on state $s$:

$$\pi(a|s) = \frac{e^{-E(s,a)}}{Z(s)} \tag{4}$$

where $Z(s)$ is a normalization factor known as the "partition function." In practice, we estimate $E$ with a learned neural network function approximator $E_\theta$ parameterized by $\theta$ and train $E_\theta$ on samples $\{s_i, a_i\}$ collected from the expert policies $\pi_H$. Training $E_\theta$ is described in Appendix 7.2.

## 4.2 Implicit Interactive Dataset Aggregation

Behavior cloning is prone to distribution shift due to compounding approximation errors [9], and any data-driven robot policy may encounter edge cases during execution that are not represented in the training data [13]. We extend IBC to interactive imitation learning using dataset aggregation of online human data, and iteratively update the shared robot policy with the aggregate dataset at a fixed interval $1 \leq \hat{t} \leq T$ via supervised learning, as in DAgger [9] and variants [11, 13]:

$$\begin{cases} D^{t+1} \leftarrow D^t \cup D_H^t, \text{ where } D_H^t := \{(s_i^t, \pi_H^j(s_i^t)) : \pi_H^j(s_i^t) \neq R \text{ and } \sum_{j=1}^M \boldsymbol{\alpha}_{ij}^t = 1\} \\ \pi_{\theta_t} \leftarrow \arg\min_\theta \mathcal{L}(\pi_\theta, D^t), \text{ if } t \equiv 0 \pmod{\hat{t}} \end{cases}$$

where $\pi_H^j(s_i^t)$ is the teleoperation action from human $j$ for robot $i$ at time $t$, and $\boldsymbol{\alpha}_{ij}^t$ is the assignment of human $j$ to robot $i$ at time $t$, as in Equation 2. Ross et al. [9] show that such a policy incurs approximation error that is linear in the time horizon rather than quadratic, as in behavior cloning.

## 4.3 Uncertainty Estimation for EBMs

While prior work computes the output variance among a bootstrapped ensemble of neural networks to estimate epistemic uncertainty [44, 12, 10], this approach is not applicable to implicit policies because multimodality results in a false positive: different ensemble members may select equally optimal actions from different modes, resulting in high variance despite high certainty. Furthermore, training and inference in EBMs are much more computationally expensive than in explicit models (Section 6), making ensembles of 5+ models impractical. Finally, inference in implicit models is nondeterministic, creating an additional source of variance that is not due to uncertainty.

The notion of ensemble disagreement can still be applied to EBMs by considering the action *distributions* at a given state rather than the single predicted actions. At states within the distribution of the human data, a bootstrapped EBM will predict action distributions that are concentrated around the human actions. However, outside of the human data distribution, the models have no reference behavior to imitate, and will likely predict different conditional action distributions due to random initialization, stochastic optimization, and bootstrapping. Accordingly, we propose bootstrapping 2 implicit policies and calculating the Jeffreys divergence $D_J$ [18] between them as a measure of how their conditional action distributions differ at a given state. Jeffreys divergence in this setting has two key properties: (1) it is symmetric, which is useful as neither bootstrapped policy is more correct than the other, and (2) it is computationally tractable for EBMs as it does not require estimating the partition function $Z(s)$ (Equation 4). To show (2), we derive the following novel identity (proof in Appendix 7.1):

**Identity 1.** *Let $E_1$ and $E_2$ be two energy-based models that respectively define distributions $\pi_1$ and $\pi_2$ according to Equation 4. Then,*

$$D_J\left(\pi_1(\cdot|s)\|\pi_2(\cdot|s)\right) = \mathbb{E}_{a\sim\pi_1(\cdot|s)}\left[E_2(s,a) - E_1(s,a)\right] + \mathbb{E}_{a\sim\pi_2(\cdot|s)}\left[E_1(s,a) - E_2(s,a)\right].$$

Crucially, the intractable partition functions do not appear in the expression due to the symmetry of Jeffreys divergence. We estimate the expectations in Identity 1 using Langevin sampling. Note that this method is not limited to the interactive IL setting and may have broad applications for any algorithms or systems that use energy-based models. We provide more intuition on the proposed metric in Figure 4 in the appendix and consider how this method may be generalized to a greater number of models in Appendix 7.3.

## 4.4 Energy-Based Allocation

To extend IBC to the IFL setting, we synthesize the Jeffreys uncertainty estimate with Fleet-DAgger [13]. Specifically, we set the Fleet-DAgger priority function $\hat{p} : (s, \pi_{\theta_t}) \rightarrow [0, \infty)$ to prioritize robots with high uncertainty, followed by robots that require a hard reset $R$. This produces a supervisor allocation policy $\omega$ with Fleet-EnsembleDAgger, the U.C. (Uncertainty-Constraint) allocation scheme in [13]. We refer to the composite approach as IIFL.

# 5 Experiments

## 5.1 Simulation Experiments: 2D Navigation

To evaluate the correctness of our implementation and provide visual intuition, we first run experiments in a 2D pointbot navigation environment. See Figure 1 for the maze environment, representative trajectories, and energy distribution plots. We consider discrete 2D states $s = (x, y) \in \mathbb{N}^2$ (the Cartesian pose of the robot) and continuous 2D actions $a = (\Delta x, \Delta y) \in [-1, 1]^2$ (relative changes in Cartesian pose). The maze has a fixed start and goal location and consists of a forked path around

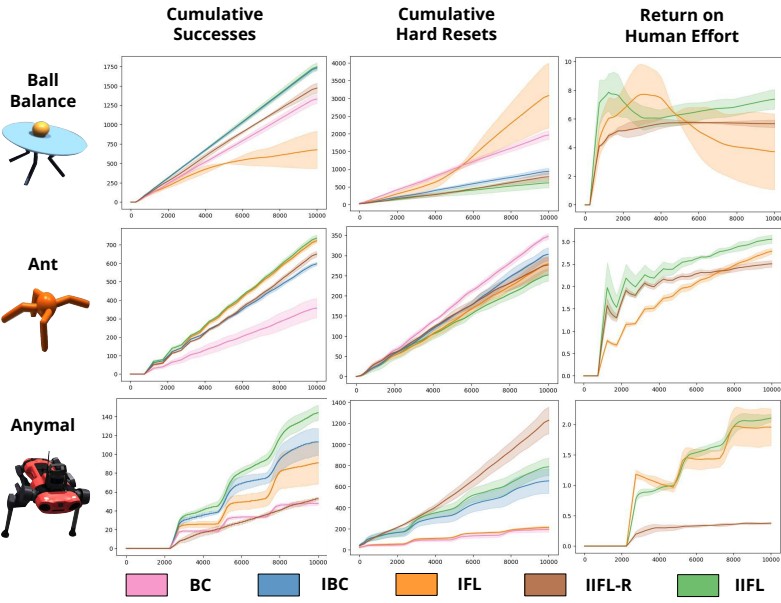

Figure 2: IFL Benchmark simulation experiment results. Despite unimodal supervision, IIFL is competitive with or outperforms IFL and other baselines across 3 environments, suggesting benefits of implicit policies beyond robustness to multimodality. Shading represents $\pm 1$ standard deviation.

a large obstacle followed by a long corridor. An algorithmic supervisor provides 100 demonstrations of the task, randomly choosing to go upward or downward at the fork with equal probability. Since a model can simply overfit to the demonstrations in this low-dimensional environment, to induce distribution shift we add "wind" at execution time to a segment of the right corridor with magnitude 0.75 in the $+y$ direction.

In 100 trials, (explicit) BC achieves a $0\%$ success rate, IBC achieves a $0\%$ success rate, and IIFL achieves a $100\%$ autonomous success rate (i.e., robot-only trajectories without human interventions, after interactive training). In Figure 1 we observe that BC cannot pass the fork due to averaging the two modes to zero. Meanwhile, IBC is not robust to the distribution shift: once the wind pushes the robot to the top of the corridor, it does not know how to return to the center. We also observe that the IIFL energy distributions in Figure 1(B) reflect the desired behavior in accordance with intuition.

### 5.2 Simulation Experiments: IFL Benchmark

**Environments:** Evaluating IIFL in simulation is uniquely challenging as it requires all of the following, precluding the use of most existing benchmarks in similar papers: (1) efficient simulation of large robot fleets, (2) simulation of multiple algorithmic humans, (3) interactive human control, and (4) heterogeneous human control, which is difficult to specify in joint space. To accommodate these requirements, following prior work [13] we evaluate with Isaac Gym [45] and the IFL Benchmark [13]. We separate these experiments into two domains: (1) homogeneous human control in 3 environments (Ball Balance, Ant, Anymal) to compare with prior IFL algorithms that assume unimodal supervision; (2) heterogeneous human control in FrankaCubeStack, the only Isaac Gym environment with Cartesian space control. More details are available in Appendix 7.4.

**Metrics:** Following prior work [13], we measure the total successful task completions across the fleet and the total number of hard resets. For interactive algorithms, we also measure the return on human effort (Equation 3) where reward is a sparse $r \in \{0, 1\}$ for task completion. Task execution is deemed successful if the robot completes its trajectory without a hard reset and reaches $95\%$ of expert human reward.

**Baselines:** We compare IIFL to the following baselines: (explicit) BC, IBC, (explicit) IFL (specifically, Fleet-EnsembleDAgger [13]), and IIFL-Random (IIFL-R), which is an ablation of IIFL that

| Algorithm | Avg. Reward | Task Successes | ROHE |
|-----------|-------------|----------------|------|
| BC | $29.27 \pm 14.05$ | $0.3 \pm 0.5$ | N/A |
| IBC | $24.96 \pm 0.83$ | $0.0 \pm 0.0$ | N/A |
| IFL | $230.39 \pm 53.41$ | $7.0 \pm 2.2$ | $2.30 \pm 0.53$ |
| IIFL-R | $166.24 \pm 28.63$ | $0.0 \pm 0.0$ | $1.66 \pm 0.29$ |
| IIFL | $\mathbf{784.26 \pm 122.41}$ | $\mathbf{26.7 \pm 4.5}$ | $\mathbf{7.84 \pm 1.22}$ |

Table 1: Execution results from the FrankaCubeStack Isaac Gym environment with 4 heterogeneous expert policies. IIFL significantly outperforms the baselines in average reward, task successes, and return on human effort.

allocates humans to robots randomly instead of using the Jeffreys uncertainty estimate. Human supervisors for BC and IBC perform only hard resets (i.e., no teleoperation) during execution.

**Experimental Setup:** We run experiments with a fleet of $N = 100$ robots and $M = 10$ algorithmic supervisors, where the supervisors are reinforcement learning agents trained with Isaac Gym's reference implementation of PPO [41]. All training runs have hard reset time $t_R = 5$ timesteps, minimum intervention time $t_T = 5$ timesteps, and fleet operation time $T = 10,000$ timesteps [13], and are averaged over 3 random seeds. The initial robot policy $\pi_{\theta_0}$ for all algorithms is initialized with behavior cloning on 10 full task demonstrations. While IFL trains at every timestep following prior work [13], the implicit interactive algorithms train at intervals of 1000 timesteps with an equivalent total amount of gradient steps for increased stability of EBM training.

FrankaCubeStack, in which a Franka arm grasps a cube and stacks it on another (see Appendix 7.4.2 for images and details), has several differences from the other 3 environments. First, since it allows Cartesian space control, we can script 4 *heterogeneous* supervisor policies with grasps corresponding to each face of the cube; the $M = 10$ supervisors are split into 4 groups, each of which has a unique policy. Second, due to the difficulty of scripting interactive experts, the online interventions take place at execution-time (i.e., the robot policy is frozen). Third, since there is no notion of catastrophic failure in the cube stacking environment, we do not report hard resets as there are none.

**Results:** The results are shown in Figure 2 and Table 1. In the homogeneous control experiments, we observe that IIFL rivals or outperforms all baselines across all metrics, with the exception of hard resets in the Anymal environment. We hypothesize that the latter results from learning more "aggressive" locomotion that makes greater progress on average but is more prone to failure. These results suggest that implicit policies may have desirable properties over explicit policies such as improved data efficiency and generalization even when multimodality is *not* present in the data, as suggested by prior work [16]. The severity of distribution shift due to compounding approximation error [9] in the homogeneous experiments roughly corresponds to the performance gap between BC and IFL (or IBC and IIFL). Surprisingly, (explicit) IFL underperforms BC in Ball Balance; we hypothesize that this is due to its frequent policy updates on a shifting low-dimensional data distribution. In the FrankaCubeStack environment, IIFL significantly outperforms the baselines across all metrics, indicating the value of implicit policies for heterogeneous supervision. The 74% performance gap between IFL and IIFL corresponds to the severity of multimodality in this environment. Only IFL and IIFL attain nontrivial success rates; while IIFL-R makes progress, it is not able to successfully stack the cube, suggesting that IIFL allocates human attention more judiciously.

### 5.3 Physical Experiments: Pushing Block to Target Point amid Obstacle

**Experimental Setup:** To evaluate IIFL in experiments with real-world human multimodality and high-dimensional state spaces, we run an image-based block-pushing task with a fleet of $N = 4$ ABB YuMi robot arms operating simultaneously and $M = 2$ human supervisors, similar to Hoque et al. [13]. See Figure 3 for the physical setup. Each robot has an identical square workspace with a small blue cube and rectangular pusher as an end effector. Unlike Hoque et al. [13], we add a square obstacle to the center of each workspace. The task for each robot is to push the cube to a goal region diametrically opposite the cube's initial position without colliding with the walls or the obstacle. Once this is achieved, the goal region is procedurally reset based on the new cube position. As described in Section 3, the role of human superivsion is to (1) teleoperate when requested and (2)

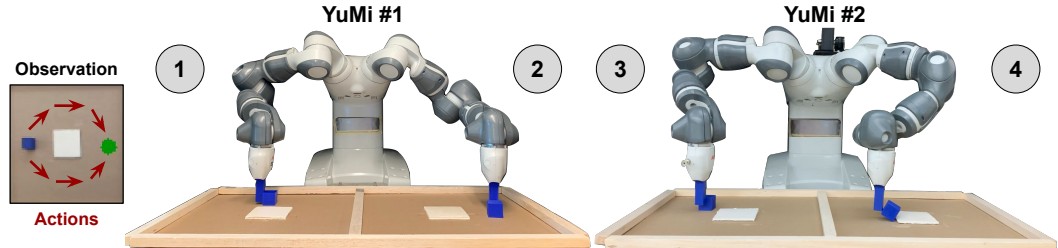

Figure 3: Physical experiment setup with 2 ABB YuMi robots for a total of 4 independent arms.

| Algorithm | Successes (↑) | Hard Resets (↓) | ROHE (↑) |
|-----------|---------------|------------------|-----------|
| BC | $2.0 \pm 0.8$ | $51.0 \pm 0.8$ | N/A |
| IBC | $20.3 \pm 4.1$ | $\mathbf{35.3 \pm 6.8}$ | N/A |
| IFL | $7.0 \pm 0.8$ | $47.3 \pm 0.5$ | $0.13 \pm 0.01$ |
| IIFL | $\mathbf{36.3 \pm 1.2}$ | $37.0 \pm 2.2$ | $\mathbf{0.71 \pm 0.01}$ |

Table 2: Physical block pushing experiment results. IIFL outperforms all baselines in number of task successes and ROHE and explicit methods in hard resets. Implicit BC and IIFL incur similar amounts of hard resets.

provide a physical hard reset when requested. When both paths to the goal are equidistant, Human 1 pushes the cube *clockwise* around the obstacle while Human 2 pushes the cube *counterclockwise*; if one path is closer, the human takes that path. Hard resets $R$ are defined to be collisions of the cube with the obstacle or the boundaries of the workspace. Furthermore, unlike the discrete action space in Hoque et al. [13], we use a continuous 2D action space of $a = (\Delta x, \Delta y)$ that corresponds to the vector along which to push the block, starting from the block's center. We run 3 trials of each algorithm in Table 2 for $T = 150$ timesteps; see Appendix 7.4.3 for more details.

**Results:** The results are shown in Table 2. We observe that implicit policies are crucial for success, as the explicit methods rarely reach the goal and incur many hard resets. Results suggest that IIFL improves the success rate by 80% over IBC and improves ROHE by $4.5\times$ over IFL. However, IIFL incurs a similar number of hard resets to IBC. We hypothesize that the duration of the physical experiment, difficult to extend due to the significant robot and human time required, is insufficient to learn subtle collision avoidance behaviors that noticeably reduce the number of hard resets.

## 6 Limitations and Future Work

Since IIFL extends IBC, it inherits some of its limitations. First, model training and inference require $18\times$ and $82\times$ more computation time than explicit methods: on one NVIDIA V100 GPU, we measure implicit training to take an additional 0.34 seconds per gradient step and implicit inference to take an additional 0.49 seconds. Second, Florence et al. [16] find that IBC performance falls on some tasks when the action space dimensionality is very high ($|\mathcal{A}| > 16$); we do not observe this in our experiments as $|\mathcal{A}| \leq 12$ but IIFL likely incurs this property with higher-dimensional actions. Third, while it is $7\times$ faster than alternate methods for implicit models and has sub-second latency for a fleet of 100 robots, IIFL uncertainty estimation is nevertheless $340\times$ slower than its highly efficient explicit counterpart (Appendix 7.4.4). Finally, the real-world evaluation of IIFL is limited to block pushing with fixed block properties; more comprehensive evaluation of IIFL in a wider range of physical domains is required to assess its full applicability.

In future work, we will evaluate IIFL in additional physical environments as well as extend recently proposed alternative approaches for handling multimodality such as Behavior Transformers [30] and Diffusion Policies [34] to the IFL setting. We will also develop algorithms that effectively learn from human demonstrations that are not only multimodal but also suboptimal. We note that as the Jeffreys uncertainty quantification method does not rely on any IFL assumptions, it may be broadly useful beyond this setting to any applications involving Boltzmann distributions and EBMs.

**Acknowledgments**

This research was performed at the AUTOLab at UC Berkeley in affiliation with the Berkeley AI Research (BAIR) Lab. The authors were supported in part by donations from Siemens, Google, Bosch, Toyota Research Institute, and Autodesk and by equipment grants from PhotoNeo, NVidia, and Intuitive Surgical. Any opinions, findings, and conclusions or recommendations expressed in this material are those of the author(s) and do not necessarily reflect the views of the sponsors. We thank our colleagues who provided helpful feedback, code, and suggestions, especially Cesar Salcedo, Letian Fu, and Aviv Adler.

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

# 7 Appendix

## 7.1 Jeffreys Divergence Identity

We derive the following identity from the main text:

**Identity 1.** *Let $E_1$ and $E_2$ be two energy-based models that respectively define distributions $\pi_1$ and $\pi_2$ according to Equation 4. Then,*

$$D_J\left(\pi_1(\cdot|s)\|\pi_2(\cdot|s)\right) = \mathbb{E}_{a\sim\pi_1(\cdot|s)}\left[E_2(s,a) - E_1(s,a)\right] + \mathbb{E}_{a\sim\pi_2(\cdot|s)}\left[E_1(s,a) - E_2(s,a)\right].$$

*Proof.* The proof follows from applying the definition of Jeffreys divergence to EBMs:

$$\begin{aligned}
D_J\left(\pi_1(\cdot|s)\|\pi_2(\cdot|s)\right) &= D_{KL}\left(\pi_1(\cdot|s)\|\pi_2(\cdot|s)\right) + D_{KL}\left(\pi_2(\cdot|s)\|\pi_1(\cdot|s)\right) \\
&= \mathbb{E}_{a\sim\pi_1(\cdot|s)}\left[\log\frac{\pi_1(a|s)}{\pi_2(a|s)}\right] + \mathbb{E}_{a\sim\pi_2(\cdot|s)}\left[\log\frac{\pi_2(a|s)}{\pi_1(a|s)}\right] \\
&= \mathbb{E}_{a\sim\pi_1(\cdot|s)}\left[E_2(s,a) - E_1(s,a)\right] - \log Z_1(s) + \log Z_2(s) \\
&\quad + \mathbb{E}_{a\sim\pi_2(\cdot|s)}\left[E_1(s,a) - E_2(s,a)\right] - \log Z_2(s) + \log Z_1(s) \\
&= \mathbb{E}_{a\sim\pi_1(\cdot|s)}\left[E_2(s,a) - E_1(s,a)\right] + \mathbb{E}_{a\sim\pi_2(\cdot|s)}\left[E_1(s,a) - E_2(s,a)\right].
\end{aligned}$$

$\square$

To provide more intuition on this identity, we plot the Jeffreys divergence for a pair of isotropic Gaussian energy functions in Figure 4.

## 7.2 Additional Details on Implicit Models

Implicit BC trains an energy-based model $E_\theta$ on samples $\{s_i, a_i\}$ collected from the expert policies $\pi_H$. After generating a set of counter-examples $\{\tilde{a}_i^j\}$ for each $s_i$, Implicit BC minimizes the following InfoNCE [46] loss function:

$$\mathcal{L} = \sum_{i=1}^{N} -\log\hat{p}_\theta(a_i|s_i, \{\tilde{a}_i^j\}), \quad \hat{p}_\theta(a_i|s_i, \{\tilde{a}_i^j\}) := \frac{e^{-E_\theta(s_i,a_i)}}{e^{-E_\theta(s_i,a_i)} + \sum_j e^{-E_\theta(s_i,\tilde{a}_i^j)}}. \tag{5}$$

This loss is equivalent to the negative log likelihood of the training data, where the partition function $Z(s)$ is estimated with the counter-examples. Florence et al. [16] propose three techniques for generating these counter-examples $\{\tilde{a}_i^j\}$ and performing inference over the learned model $E_\theta$; we choose gradient-based Langevin sampling [47] with an additional gradient penalty loss for training in this work as Florence et al. [16] demonstrate that it scales with action dimensionality better than the alternate methods. This is a Markov Chain Monte Carlo (MCMC) method with stochastic gradient Langevin dynamics. More details are available in Appendix B.3 of Florence et al. [16].

We use the following hyperparameters for implicit model training and inference:

| Hyperparameter | Value |
|---|---|
| learning rate | 0.0005 |
| learning rate decay | 0.99 |
| learning rate decay steps | 100 |
| train counter examples | 8 |
| langevin iterations | 100 |
| langevin learning rate init. | 0.1 |
| langevin learning rate final | 1e-5 |
| langevin polynomial decay power | 2 |
| inference counter examples | 512 |

Table 3: Implicit model hyperparameters.

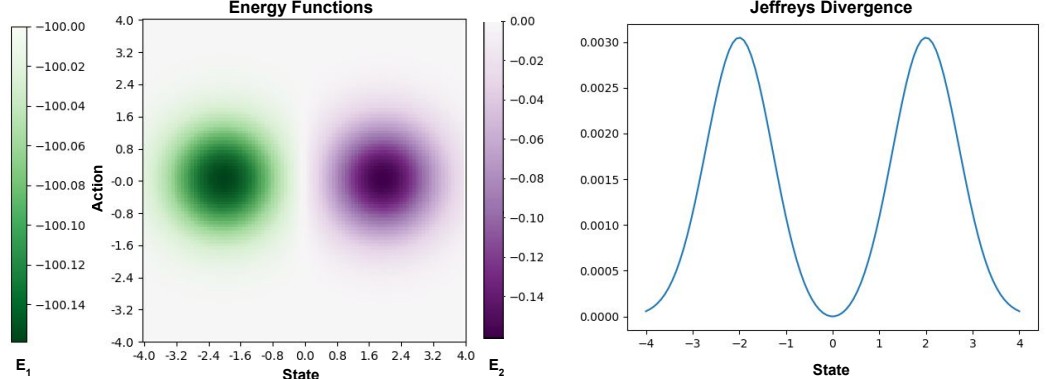

Figure 4: Consider a pair of isotropic Gaussian energy functions $E_1(s, a)$ and $E_2(s, a)$ in green and purple respectively, where each function is a negated Gaussian probability density function and $E_1$ adds a uniform offset of $Z = -100$ to all values (Left). Using numerical integration to directly compute the expectations in the Jeffreys divergence identity (Identity 1), at each state we calculate the distance between the implicit policies defined by the two energy functions (Right). As intuition suggests, the divergence peaks at the mean of each Gaussian (where one energy function is highest and the other is near zero) and approaches zero where the energy functions are the same (at the center and edges of the state space). Note the symmetric structure of the Jeffreys curve, which produces identical values regardless of the offset $Z$.

## 7.3 Uncertainty Estimation with Larger Ensembles

Prior works using ensembles of explicit models to estimate epistemic uncertainty [44, 12, 10] typically employ larger ensembles of $n \geq 5$ models, whereas IIFL uses $n = 2$. We wish to evaluate the impact of this smaller number of models. However, the Jeffreys divergence is only defined for two distributions, and while other divergence measures (e.g. Jensen-Shannon) can be generalized to an arbitrary number of distributions, they typically require knowledge of the intractable partition functions of the distributions. Accordingly, we consider estimating the uncertainty of $n = 5$ implicit models by computing the average of the Jeffreys divergences between every pairwise combination of models. Figure 5 provides intuition on this measure, and we provide information on computation time in Section 7.4.4.

We evaluate the effect of adding more models by comparing the estimate of the Jeffreys divergence with $n = 2$ models and the averaged estimate with $n = 5$ models to the L2 distance between the robot policy's proposed action and the expert policy's action at the same state. While ground truth epistemic uncertainty is intractable to calculate, the ground truth action discrepancy between the human and robot can provide a correlate of uncertainty: higher discrepancy corresponds to higher uncertainty. The results are shown in Figure 6. We observe that both ensemble sizes are positively correlated with action discrepancy, and that the ensemble with $n = 5$ models has a higher correlation ($r = 0.804$) than the ensemble with $n = 2$ models ($r = 0.688$). We also observe that the $n = 5$ ensemble has lower variance than $n = 2$: the standard deviation is 0.176 compared to 0.220. These results suggest that larger ensembles can improve the uncertainty estimation at the cost of increased computation time ($2.6\times$ in Section 7.4.4).

## 7.4 Additional Experimental Details

### 7.4.1 IFL Benchmark Hyperparameters

Implementations of Implicit Interactive Fleet Learning and baselines are available in the code supplement and are configured to run with the same hyperparameters we used in the experiments. To compute the uncertainty thresholds $\hat{u}$ for Explicit IFL and IIFL (see Section 8.3.1 in [13] for definition), we run Explicit BC and Implicit BC respectively with $N = 100$ robots for $T = 1000$ timesteps and choose the 99th percentile value among all $100 \times 1000$ uncertainty values. The FrankaCubeStack environment sets these thresholds to zero since there are no constraint violations (i.e., this sorts robot

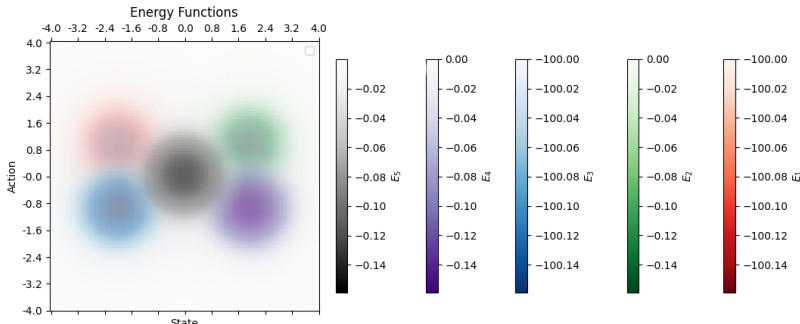

(a) Consider 5 isotropic Gaussian energy functions, each a negative Gaussian probability density function with some offset.

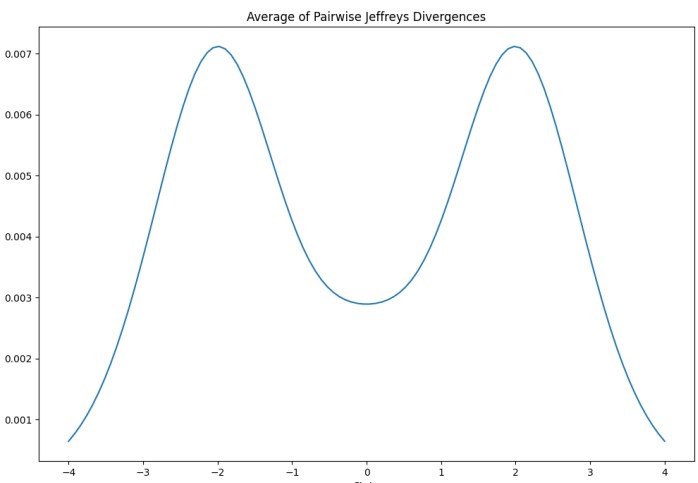

(b) We use numerical integration to calculate at each state the Jeffreys divergences between each of the $\binom{5}{2} = 10$ unique pairs of models, and report the average value. As intuition suggests, the calculated uncertainty is highest at states $-2$ and $2$, where two of the Gaussians have means that are far apart, meaning that they strongly prefer very different actions. At state $0$, the uncertainty is lower as one model strongly prefers the action $0$, and the others are closer to uniform. Far from state $0$, the uncertainty is lowest as all the energy functions are approximately flat.

Figure 5

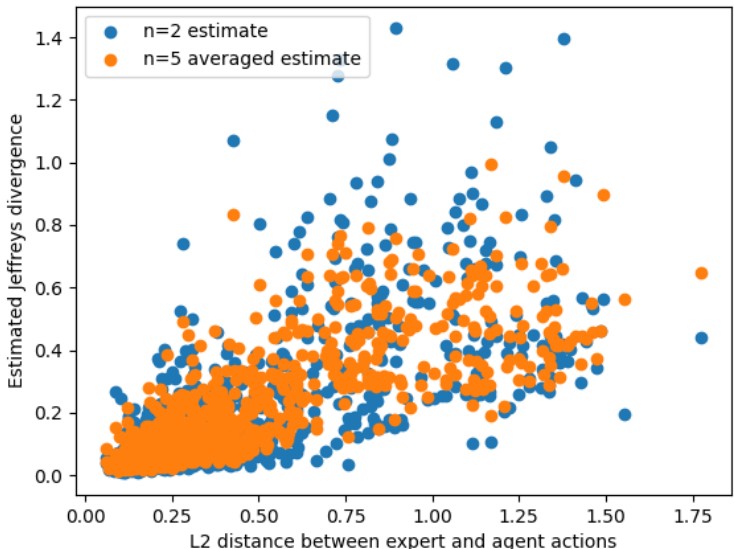

Figure 6: We plot the Jeffreys divergence estimates and the ground truth action discrepancies at the first 1000 states visited by a robot with a unimodal policy. Both variants of the Jeffreys divergence calculation are positively correlated with the $L2$ distance between the robot policy's and expert policy's actions. In the $n = 2$ case, the correlation coefficient is $r = 0.688$; in the $n = 5$ case, the correlation coefficient is $r = 0.804$, indicating that additional models can make the ensemble more predictive of when the agent will deviate from the expert (at the cost of increased computation time).

priority by uncertainty alone). See Table 4 for these values, state and action space dimensionality, and other hyperparameters. The batch size is 512 and all algorithms pretrain the policy for $N/2$ gradient steps, where $N$ is the number of data points in the 10 offline task demonstrations. Finally, as in prior work [13], the Random IIFL baseline is given a human action budget that approximately equals the average amount of human supervision solicited by IIFL. See the code for more details.

| Environment | $|S|$ | $|A|$ | Explicit $\hat{u}$ | Implicit $\hat{u}$ |
|---|---|---|---|---|
| BallBalance | 24 | 3 | 0.1179 | 0.1206 |
| Ant | 60 | 8 | 0.0304 | 0.9062 |
| Anymal | 48 | 12 | 0.0703 | 2.2845 |
| FrankaCubeStack | 19 | 7 | 0.0 | 0.0 |

Table 4: Simulation environment hyperparameters.

### 7.4.2 FrankaCubeStack Environment

The scripted supervisor for FrankaCubeStack is defined in `human_action()` of `env/isaacgym/franka_cube_stack.py` in the code supplement. Using known pose information and Cartesian space control, the supervisor policy does the following, where Cube A is to be stacked on Cube B: (1) move the end effector to a position above Cube A; (2) rotate into a pre-grasp pose; (3) descend to Cube A; (4) lift Cube A; (5) translate to a position above Cube B; (6) place Cube A on Cube B; and (7) release the gripper. Heterogeneity is concentrated in Step 2: while one supervisor rotates to an angle $\theta \in [0, \frac{\pi}{2}]$ that corresponds to a pair of antipodal faces of the cube, the others rotate to $\theta - \pi, \theta - \frac{\pi}{2}$, and $\theta + \frac{\pi}{2}$. See Figure 7 for intuition. We also include results for only 2 hetereogeneous policies ($\theta$ and $\theta - \frac{\pi}{2}$) in Table 5; results (in conjunction with Table 1) suggest that relative performance of IIFL over baselines remains approximately consistent as the number of modes varies and can improve as multimodality increases.

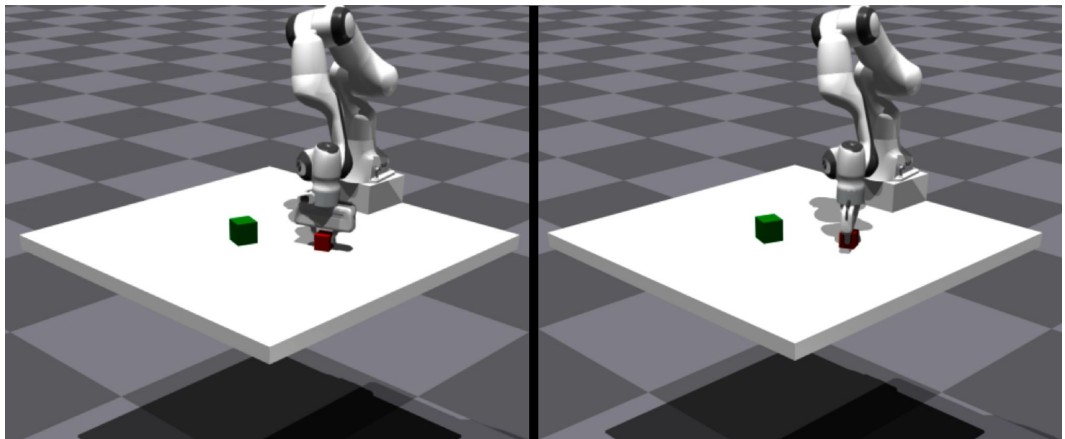

Figure 7: The scripted heterogeneous supervisors for the FrankaCubeStack Isaac Gym environment pick different faces of the cube for the same cube pose.

| Algorithm | Avg. Reward | Task Successes | ROHE |
|---|---|---|---|
| BC | $23.45 \pm 0.99$ | $0.0 \pm 0.0$ | N/A |
| IBC | $30.32 \pm 2.78$ | $0.0 \pm 0.0$ | N/A |
| IFL | $307.87 \pm 118.59$ | $9.3 \pm 4.7$ | $3.08 \pm 1.19$ |
| IIFL-R | $244.98 \pm 32.58$ | $0.0 \pm 0.0$ | $2.45 \pm 0.33$ |
| IIFL | $\mathbf{604.17 \pm 263.06}$ | $\mathbf{17.7 \pm 11.1}$ | $\mathbf{6.04 \pm 2.63}$ |

Table 5: Execution results from the FrankaCubeStack Isaac Gym environment with 2 heterogeneous supervisor policies (rather than 4).

### 7.4.3 Physical Experiment Protocol

We largely follow the physical experiment protocol in Hoque et al. [13] but introduce some modifications to human supervision. We execute 3 trials of each of 4 algorithms (Explicit BC, Implicit BC, Explicit IFL, Implicit IFL) on the fleet of 4 robot arms. Each trial lasts 150 timesteps (synchronous across the fleet) for a total of $3 \times 4 \times 4 \times 150 = 7200$ individual pushing actions. The authors provide human teleoperation and hard resets, which differ from prior work due to the continuous action space and the square obstacle in the center of the workspace. Teleoperation is done using an OpenCV (https://opencv.org/) GUI by clicking on the desired end point of the end-effector in the overhead camera view. Hard resets are physical adjustments of the cube to a randomly chosen side of the obstacle. IIFL is trained online with updated data at $t = 50$ and $t = 100$ while IFL is updated at every timestep (with an equivalent total amount of gradient steps) to follow prior work [13].

The rest of the experiment protocol matches Hoque et al. [13]. The 2 ABB YuMi robots are located about 1 km apart; a driver program uses the Secure Shell Protocol (SSH) to connect to a machine that is connected to the robot via Ethernet, sending actions and receiving camera observations. Pushing actions are executed concurrently by all 4 arms using multiprocessing. We set minimum intervention time $t_T = 3$ and hard reset time $t_R = 5$. All policies are initialized with an offline dataset of 3360 image-action pairs (336 samples collected by the authors with $10\times$ data augmentation). $10\times$ data augmentation on the initial offline dataset as well as the online data collected during execution applies the following transformations:

- Linear contrast uniformly sampled between 85% and 115%

- Add values uniformly sampled between -10 and 10 to each pixel value per channel

- Gamma contrast uniformly sampled between 90% and 110%

- Gaussian blur with $\sigma$ uniformly sampled between 0.0 and 0.3

- Saturation uniformly sampled between 95% and 105%

- Additive Gaussian noise with $\sigma$ uniformly sampled between 0 and $\frac{1}{80} \times 255$ $80 \times 255$

### 7.4.4  Computation Time

In Table 6 we report the mean and standard deviation of various computation time metrics. All timing experiments were performed with $N = 100$ robots and averaged across $T = 100$ timesteps in the Ant environment on a single NVIDIA Tesla V100 GPU with 32 GB RAM. Training time is reported for a single gradient step with a batch size of 512. Note that with default hyperparameters, IFL trains an ensemble of 5 (explicit) models and IIFL trains an ensemble of 2 (implicit) models; hence, we also report the training time per individual model. IFL inference consists of a single forward pass through each of the 5 models, while IIFL inference performs 100 Langevin iterations; both of these are vectorized across all 100 robots at once. IFL uncertainty estimation also consists of a single forward pass through each of the 5 models while IIFL performs both Langevin iterations and 2 forward passes through each of the 2 models. While IIFL can provide policy performance benefits over IFL, we observe that it comes with a tradeoff of computation time, which may be mitigated with parallelization across additional GPUs. Furthermore, while uncertainty estimation is the bottleneck in IIFL, it is performed with sub-second latency for the entire fleet. This is significantly faster than alternatives such as directly estimating the partition function, which is both less accurate and slower; we measure it to take an average of 7.10 seconds per step using annealed importance sampling [48]. Finally, uncertainty estimation for the variant described in Section 7.3 that uses $n = 5$ implicit models required 2.599±0.002s. While the time complexity should grow as quadratic in $n$, in practice we observe that for small values of $n$ the growth is closer to linear as the latency is dominated by the $\mathcal{O}(n)$ sampling process rather than the $\mathcal{O}(n^2)$ forward passes.

| Time | IFL | IIFL |
|---|---|---|
| Training step (s) | $0.0385 \pm 0.0205$ | $0.694 \pm 0.207$ |
| Training step per model (s) | $0.0077 \pm 0.0041$ | $0.347 \pm 0.104$ |
| Inference (s) | $0.0060 \pm 0.0395$ | $0.494 \pm 0.045$ |
| Uncertainty estimation (s) | $0.0029 \pm 0.0008$ | $0.988 \pm 0.008$ |

Table 6: Computation times for training, inference, and uncertainty estimation for IFL and IIFL.

