# OpenReview forum: "IIFL: Implicit Interactive Fleet Learning from Heterogeneous Human Supervisors"
_robot-learning.org/CoRL/2023/Conference — CoRL 2023 Poster_

### Official Review · Reviewer_9GiX · 2023-07-16

**Confidence:** 4
**Originality:** Good
**Technical Quality:** Very Good
**Clarity Of Presentation:** Very Good
**Impact:** 3

**Recommendation:**

Strong Accept: I recommend accepting the paper and will argue for my recommendation even if other reviewers hold a different opinion.

**Review:**

Strengths:
- The problem of interactive learning from human demonstrations that often can be multi-modal is important for scaling up robot learning and deploying robots for an effective data collection.
- Using Jeffreys divergence is a particularly interesting way to estimate epistemic uncertainty and can be used for a wide range of tasks.
- The paper is well-written and easy to understand and follow.
- The method is shown to outperform baselines in simulation and also work in the real world on multiple robotic arms.

Weaknesses:
- In the paper, the method uses 2 implicit models per policy to compute uncertainty. It would be useful to see an ablation on whether more models can improve this estimation (at a cost of increased compute requirements) as it is not completely clear how well uncertainty can be estimated from such a small amount of models.
- Generally it would be interesting to see a wider range of real world experiments on more articulated tasks with a wider variety of objects to better understand the general applicability of the method and how well implicit models can estimate uncertainty in more complex environments.

**Quality Of The Limitations Section:**

Limitations are addressed clearly

**Questions For Rebuttal:**

Please refer to the review for rebuttal questions.

**Robotics Focus:**

Sufficient demonstration on hardware

**Summary Of Paper:**

The paper presents a method for interactive learning of robotic policies from multi-modal human demonstrations. It uses implicit behavior cloning to represent a learned policy as an energy model, where actions can be extracted by optimizing over it. As the policies are learned on a fleet of multiple robots, the interactive part is based on uncertainty prioritized DAgger-style method where a robot with the highest uncertainty is chosen for humans to provide demonstrations. The paper introduces a novel way to estimate epistemic uncertainty by using the Jeffreys divergence metric to estimate the distance between two distributions from two energy models. The method is tested on simulated tasks from the Isaac Gyms suite and a real world block pushing task on 4 robotic arms from 2 ABB Yumi robots. The method is shown to outperform non-interactive baselines, such as Behavioral Cloning (BC) and Implicit BC, and Interactive Fleet Learning (IFL) baseline that did not use implicit policies.

**Summary Of Recommendation:**

The paper introduces a novel way to estimate uncertainty with implicit policy models and deploys it for a an interactive data collection on multiple agents. The paper is interesting but some more experiments on a wider range of real world tasks would help better understand the general applicability of the method.

UPDATE after rebuttal:
Thank you for new experiments, I increased my score.

---

### Official Review · Reviewer_XAiu · 2023-07-19

**Confidence:** 3
**Originality:** Fair
**Technical Quality:** Good
**Clarity Of Presentation:** Very Good
**Impact:** 3

**Recommendation:**

Weak Accept: I recommend accepting the paper, but will not argue for my recommendation if the majority of other reviewers have a different opinion.

**Review:**

The idea of combining two existing approaches (implicit policies and interactive learning) as a mechanism to address two of the key challenges in imitation learning is interesting. The paper is well-written, the related work does a good job of motivating the proposed idea and its relevance to robotic applications, and the mathematical exposition is clear and well-presented. The inclusion of a physical robot experiment as part of the evaluation is also greatly appreciated.

The main weakness of the paper in its current state relates to the “Experimental Results” section. Specifically, although the chosen experiments try to showcase how IIFL can mitigate the effects of distribution shift and multimodality, the current description does not provide clear details on how severe these two issues are nor how IIFL scales to more complex setups.



**Quality Of The Limitations Section:**

Additional details required

**Questions For Rebuttal:**

For instance, in the case of the ball balance, ant and animal benchmark, how are distribution shifts introduced during testing? Also, it is not clear why IIGL results in a higher number of hard resets for the Anymal environment.

Experiments focused on showcasing how IIFL can handle multimodality are limited to 2 possible demonstrators. It would be interesting to show how IIFL's performance changes as the number of heterogeneous supervisors increases (using the FrankaCubeStack task for example).

**Robotics Focus:**

Sufficient demonstration on hardware

**Summary Of Paper:**

This paper presents implicit interactive fleet learning (IIFL), an approach that mitigates the impact of distribution shift and multimodality on robot policies learned via imitation learning. The general idea is to combine implicit policies with interactive fleet learning (a state-of-the-art approach where multiple supervisors are available to provide corrective feedback during learning). While the implicit policies allow for heterogeneous supervision (multimodality), interactive learning helps to mitigate distribution shifts. The paper also proposes to use Jeffrey's uncertainty estimate as the criterion to determine which robots should be first assigned to a human supervisor. Experiments in simulation and with a physical robot show that IIFL results in improved robot performance (success) and return on human effort.


**Summary Of Recommendation:**

This is an interesting paper with a potential impact on learning more robust and diverse robot policies from human demonstrations, however, the current evaluation provides limited evidence and few details on how much multimodality and/or distributional shift can be handled by the proposed approach before performance starts to degrade.

---

### Official Review · Reviewer_z8az · 2023-07-20

**Confidence:** 5
**Originality:** Good
**Technical Quality:** Good
**Clarity Of Presentation:** Very Good
**Impact:** 3

**Recommendation:**

Weak Accept: I recommend accepting the paper, but will not argue for my recommendation if the majority of other reviewers have a different opinion.

**Review:**

Strengths:
1. The paper is well-written and provides a clear description of the setup and implementation details.
2. The novel integration of implicit Behavioral Cloning (BC) and human-in-the-loop imitation learning presents a fresh perspective. The usage of symmetric KL divergence to facilitate an implicit policy that adapts to human supervision is straightforward and neat.
3. The experimental results demonstrate significant performance enhancements, with an impressive 80% improvement in real-world robot tasks.

Weaknesses:
1. While I appreciate the endeavor, my main concern with this work lies in the handling of multimodality and distribution shift, which, in my opinion, represent two orthogonal challenges for imitation learning. While it's reasonable to fuse these elements for optimal imitation learning performance, it resembles more of an A+B solution with restricted novelty. A more natural focus could be on interactive learning from multiple human supervisors, as different supervisors may offer varied feedback - hence the necessity for an implicit model to manage such multimodal feedback. If the multimodality in this work stems from task design, offering multiple solutions to the goal as indicated in Figures 1 and 4, then it seems to be a subject independent of interactive learning.
2. The experimental outcomes presented in the first image of Figure 3 appear weird, suggesting that the method with interactive human feedback underperforms the BC baseline. This could imply that interactive learning might not always enhance imitation learning. It would be beneficial for the authors to provide an explanation for this outcome, elaborating on potential underlying factors.
3. The experiments are mostly ablation studies and lacks comprehensive comparisons with recent works in interactive learning, such as reference [22].
4. The real-world experiments seem overly simplified. The task design fails to showcase the true potential of interactive imitation learning, given that successful pushing results using imitation learning can be achieved without the necessity for interactive learning as is shown in many prior works.
5. I like the idea of employing symmetric KL divergence to fine-tune the implicit model based on human feedback. However, how does the current approach deal with the sparsity lies in the human feedbacks? The existing method seems to rely on an assumption that dense and consistent human feedback will be available to shape the distribution. It might be beneficial to explore ways to increase sample-efficiency and manage sparse feedback that encompasses diverse corner cases, especially in an interactive learning setting with multiple supervisors.
6. While the proposed method has the capacity to manage multiple human supervisors, the experiments limits itself to just two. It raises the question whether the inclusion of more supervisors might present any complications to the algorithm.

**Quality Of The Limitations Section:**

Limitations are addressed clearly

**Questions For Rebuttal:**

1. Given the distinct nature of multimodality and distribution shift in imitation learning, could you elaborate on the decision to combine both elements in the IIFL approach and its impact on the novelty of the solution?
2. How does the IIFL approach tackle the sparsity of human feedback, given that its current design seems to rely on the availability of dense and consistent feedback to form the distribution?

**Robotics Focus:**

Sufficient demonstration on hardware

**Summary Of Paper:**

This work introduces Implicit Interactive Fleet Learning (IIFL), a novel imitation learning algorithm that addresses both distribution shift and multimodality issues, common in conventional methods. IIFL integrates implicit models with interactive fleet learning and applies Jeffreys divergence to estimate uncertainty in energy-based models. The method demonstrates impressive performance, delivering a 4.5x higher return on human effort in simulation experiments and an 80% higher success rate in a physical block-pushing task.

**Summary Of Recommendation:**

In conclusion, while the paper offers valuable insights into addressing multimodality and distribution shift in imitation learning, I propose a weak rejection at this stage due to my concerns about the orthogonality of these problems and the perceived limited novelty of the A+B approach. However, I am open to reconsidering my rating based on the authors' replies.

---

### Author Response · Authors · 2023-08-10
**Rebuttal Posted**

We thank the reviewers for their detailed reviews and helpful suggestions for improving our submission. We have responded to each reviewer’s concerns in the comment replies below and attached a revision of our draft to each rebuttal with changes highlighted in blue. Major updates include: (1) new experimental results with 4 heterogeneous policies (rather than 2) in the FrankaCubeStack environment as Reviewer XAiu suggests; (2) changes to the paper framing to emphasize multi-human interactive learning as Reviewer z8az suggests; (3) new analysis of increased ensemble size as Reviewer 9GiX suggests.

---

### Author Response · Authors · 2023-08-15
**Thanks**

As this phase of the rebuttal period comes to a close, thanks again to all reviewers for their helpful suggestions and responsiveness. We are glad we were able to address the concerns raised and improve the manuscript as a result.

---

### Decision · Program_Chairs · 2023-08-30

**Decision:**

Accept (Poster)

**Comment:**

This work introduced an implicit interactive fleet learning method, combining implicit policy learning into the interactive fleet learning framework. It aims to address the multi-human multimodality and the distribution shift when the robot fleet learns from multiple human supervisors. A key innovation behind this work is using Jeffrey's uncertainty estimate to determine which robots should be first assigned to a human supervisor. At the initial reviews, some clarification questions and concerns about the novelty were raised. The authors did a great job in their rebuttal, addressing most of the issues brought up by the reviewers. Toward the end of the discussion period, all three reviewers felt positive about this work. The AC agreed with Reviewer z8az that "The real-world experiments seem overly simplified." This work could be made much stronger if more complex tasks had been evaluated with the physical robot. Nonetheless, this work has offered a convincing solution to an important problem in interactive learning. The AC supports the reviewers' recommendations to accept this paper at CoRL.